# A comparison between in situ monazite Lu–Hf and U–Pb geochronology

Alexander T. De Vries Van Leeuwen[1,2,3], Stijn Glorie[1,2], Martin Hand[1,2], Jacob Mulder[1], Sarah E. Gilbert[4]

[1]Department of Earth Sciences, University of Adelaide, Adelaide, SA, Australia
[2]Mineral Exploration Cooperative Research Centre, Kensington, WA, Australia
[3]Department of Energy and Mining, Geological Survey of South Australia, Adelaide, SA, Australia
[4]Adelaide Microscopy, University of Adelaide, Adelaide, SA, Australia

*Correspondence to*: Alexander T. De Vries Van Leeuwen (alexander.devriesvanleeuwen@adelaide.edu.au)

**Abstract.** In complex metamorphic rocks, monazite U–Pb dates can span a wide concordant range, leading to ambiguous geological interpretations (e.g., slow protracted cooling versus multiphase growth). We present in situ monazite Lu–Hf analysis as an independent chronometer to verify U–Pb age interpretations. Monazite Lu–Hf dates were attained via laser ablation inductively coupled plasma mass spectrometry equipped with collision/reaction cell technology (LA-ICP-MS/MS). In situ Lu–Hf dates for potential reference monazites with uncertainties < 1.6% agree with published U–Pb dates, validating the approach. We demonstrate the method on complex metamorphic samples from the Arkaroola region of the northern Flinders Ranges, South Australia, which exhibit protracted thermal and monazite growth histories due to high geothermal gradient metamorphism. In situ Lu–Hf dates reproduce the main U–Pb monazite age populations, demonstrating the ability to reliably resolve multiple age populations from polymetamorphic monazite samples.

## 1 Introduction

Monazite is a common accessory mineral in a broad range of metamorphic and felsic igneous rocks and forms across wide-ranging pressure–temperature conditions. In metamorphic rocks, monazite can record multiple stages of crystal growth (e.g., Kohn and Malloy, 2004; Rubatto et al., 2013), undergo fluid-mediated dissolution-precipitation reactions (e.g., Harlov et al., 2011; Seydoux-Guillaume et al., 2002), and at high temperatures and/or strain rates undergo recrystallisation (e.g., Erickson et al., 2015; Kelly et al., 2012). This responsiveness to changing physicochemical conditions makes monazite amenable to recording multiple overprinting events and complex episodes of fluid-rock interaction. Consequently, U–Pb dating of monazite has become routine for deciphering the timing and tempo of thermal events in crustal rocks (e.g., Kohn and Malloy, 2004; Larson et al., 2022; Parrish, 1990; Rubatto et al., 2001).

Widely dispersed concordant dates are a common observation in monazite U–Pb data from metamorphic rocks in complex and/or long-lived orogens (e.g., Clark et al., 2024; De Vries Van Leeuwen et al., 2021; Kirkland et al., 2016; Korhonen et al., 2013). There is often ambiguity surrounding what this dispersion represents. Common interpretations consider prolonged, slow

cooling and associated volume diffusion, or partial dissolution-reprecipitation by overprinting or prolonged thermal events. Detailed microstructural observations and trace element geochemistry play a key role in contextualising these data, however, in their absence or ambiguity, the significance of dispersion in U–Pb dates can be difficult to interpret. As such, it is important to understand the significance of such concordia dispersion, as it can lead to substantially different tectonic interpretations.

With the recent advent of in situ Lu–Hf dating facilitated by LA-ICP-MS/MS, a new frontier of in situ dating opportunities has emerged (e.g., Glorie et al., 2023; Simpson et al., 2021, 2022; Yu et al., 2024). In this contribution, we first appraise in situ Lu–Hf isotopic data from monazite reference materials and in-house secondary reference materials by comparing calculated Lu–Hf dates with published U–Pb dates. We then compare the results of in situ Lu–Hf and U–Pb geochronology from monazites that record a protracted history of fluid-driven dissolution and re-precipitation. Monazite Lu–Hf dating by LA-ICP-MS/MS was recently demonstrated using an iCap TQ instrument (Wu et al., 2024). However, this instrumental approach lacks axial ion acceleration and the ability to set a wait time between isotope jumps. These limitations affect sensitivity and induce undesirable isobaric interferences, hindering the exploration of in situ monazite Lu–Hf dating to its full potential. Here we present monazite Lu–Hf data acquired using an Agilent 8900x mass-spectrometer, with demonstrated better performance for heavy ions, and show that even in complex systems with protracted thermal histories, monazite Lu–Hf dating yields robust geochronometric data that can be used to interrogate U–Pb dates. In situ Lu–Hf dating of monazite can resolve multiple age populations from single grains and thus may be useful in cases where the U–Pb system has been compromised by Pb-loss, non-radiogenic Pb contamination, excess $^{206}$Pb due to $^{230}$Th uptake, low U concentration, or a combination of these factors.

## 2 Methods

### 2.1 Lu–Hf geochronology and trace element geochemistry

Monazite Lu–Hf geochronological and trace element analysis was undertaken at Adelaide Microscopy, at The University of Adelaide, following Simpson et al. (2021), which we briefly outline here. Analyses of Lu–Hf were acquired across two sessions using a RESOlution-LR 193 nm excimer laser ablation system coupled to an Agilent 8900 ICP-MS/MS. The reaction gas used was $NH_3$, supplied as a mixture of 10% $NH_3$ in 90% He. Laser beam diameters were set to either 43 or 67 µm, depending on Lu concentrations and microstructural constraints (e.g., size and shape of monazite compositional domains). The laser repetition rate was 10 Hz with an average on-sample fluence of ~3.5 J cm$^{-2}$. The ablated sample material was transported from the laser cell to the ICP-MS by a He carrier gas (380 mL min$^{-1}$). Data acquisition consisted of: (1) 30 seconds of baseline acquisition; (2) 40 seconds of continuous ablation, during which data were collected; and (3) ~25 seconds of washout. The following isotopes (mass shifts denoted in parentheses) were measured: $^{27}$Al, $^{43}$Ca, $^{(47+66)}$Ti, $^{88}$Sr, $^{(89+83)}$Y, $^{(90+83)}$Zr, $^{139}$La, $^{140}$Ce, $^{141}$Pr, $^{146}$Nd, $^{147}$Sm, $^{153}$Eu, $^{157}$Gd, $^{159}$Tb, $^{163}$Dy, $^{165}$Ho, $^{166}$Er, $^{169}$Tm, $^{172}$Yb, $^{175}$Lu, $^{(175+82)}$Lu, $^{(176+82)}$Hf, $^{(178+82)}$Hf, and $^{(232+15)}$Th. Axial acceleration was set to 2 V and a wait time offset of 2 ms was set to avoid memory effects when cycling between isotopes. $^{175}$Lu was measured as a proxy for $^{176}$Lu and $^{178}$Hf as a proxy for $^{177}$Hf. The calculation of $^{176}$Lu and $^{177}$Hf was

performed assuming present-day $^{176}Lu/^{175}Lu$ (0.02659) and $^{177}Hf/^{178}Hf$ (0.682) ratios (De Biévre and Taylor, 1993), following the procedures outlined in Simpson et al. (2021). Isobaric interference from $^{(176+82)}Lu$ on $^{(176+82)}Hf$ was corrected by monitoring $^{(175+82)}Lu$ and subtracting a proportion of this signal from $^{(176+82)}Hf$ based on the present-day $^{176}Lu/^{175}Lu$ ratio. No corrections were performed for isobaric interferences from $^{(176+82)}Yb$ on $^{(176+82)}Hf$, as this has been demonstrated to be negligible (Simpson et al., 2021).


Data reduction was performed in LADR (Norris and Danyushevsky, 2018). Background-subtracted isotopic ratios were normalised to NIST 610 glass using the Nebel et al. (2009) isotope dilution multi-collector inductively coupled plasma mass spectrometry (ID-MC-ICP-MS) isotopic compositions of $^{176}Lu/^{177}Hf = 0.1379 \pm 0.005$ and $^{176}Hf/^{177}Hf = 0.282111 \pm 0.000009$. Analyses of NIST 610 were conducted before and after every 40 unknown analyses and were also used to normalise isotopic

ratios and correct for instrument drift. No downhole fractionation corrections were applied, as there was no observable downhole fractionation. This is consistent with the results of Simpson et al. (2021), where no downhole fractionation was observed in garnet, apatite, or xenotime, using laser beam diameters between 43 µm and 120 µm. A subsequent matrix fractionation correction was applied to the calculated $^{177}Lu/^{176}Hf$ ratios (cf. Simpson et al., 2021, 2023). Although matrix-matched reference materials are desirable, it has been demonstrated that when using the same method and instrumentation as

that outlined in Simpson et al. (2021), correction factors for materials with similar ablation characteristics analysed with the same laser beam conditions are indistinguishable (e.g., Glorie et al., 2023, 2024a). Here, the Bamble-1 and OD-306 apatite reference materials were employed to perform matrix fractionation corrections for sessions 1 and 2, respectively. Bamble-1 ($1102 \pm 5$ Ma; Simpson et al., 2024) yielded an uncorrected inverse Lu–Hf isochron age of $1150 \pm 8$ Ma ($n = 20$, MSWD = 2.0, $p = 0.00$) and OD-306 ($1597 \pm 7$ Ma; Thompson et al., 2016) yielded an uncorrected inverse Lu–Hf isochron age of 1671

$\pm 15$ Ma ($n = 25$, MSWD = 0.94, $p = 0.55$). This resulted in correction factors of $4.40 \pm 0.04$ % and $4.71 \pm 0.05$ % for sessions 1 and 2, respectively. Monazite reference materials TS-Mnz (Budzyń et al., 2021) and RW-1 (Ling et al., 2017) were analysed in both sessions to appraise the accuracy of these corrections (discussed below). Additionally, in-house apatite secondary reference material HR-1 (long-term Lu–Hf age of $344 \pm 2$ Ma; Glorie et al., 2024a) was analysed across both sessions and yielded corrected isochron ages of $348 \pm 4$ Ma ($n = 15$, MSWD = 1.30, $p = 0.17$) and $342 \pm 3$ Ma ($n = 26$, MSWD = 0.96, $p =$

0.52) for sessions 1 and 2, respectively. Trace element data were calibrated using NIST 610, with Ce used as the internal standardisation element for calibration. Ce concentrations were set to 21.39 wt% for RW-1, 21.42 wt% for TS-Mnz, and 20 wt% for all other samples. Trace element concentrations were quantified by normalising wt% oxides to 100% totals.

Inverse Lu–Hf isochron and weighted mean ages were calculated using IsoplotR (Vermeesch, 2018), using a $^{176}Lu$ decay

constant of $0.00001867 \pm 0.00000008$ Myr$^{-1}$ (Söderlund et al., 2004). Given the narrow range of initial terrestrial $^{177}Hf/^{176}Hf$ ratios, anchored regressions were used to calculate inverse isochrons (following the approach of Glorie et al., 2024a, b). All inverse isochron plots presented in this study were anchored to an initial $^{177}Hf/^{176}Hf$ value of $3.55 \pm 0.05$, covering the range of plausible terrestrial possibilities (Spencer et al., 2020). This avoids issues which may be encountered when calculating

regressions on samples with low $^{177}$Hf/$^{176}$Hf variability, as samples with little spread along the isochron can lead to spurious

upper intercepts yielding geologically implausible initial Hf values (e.g., Vermeesch, 2024). The algorithm employed for performing anchored regressions is detailed in Vermeesch et al. (2024). Both analytical and propagated uncertainties are presented following the format: $t \pm x$ [$y$] Ma, where $t$ = the calculated Lu–Hf date, $x$ = the analytical 2SE uncertainty, and $y$ = the propagated uncertainty. Error propagation involved the quadratic addition of uncertainties on the measured sample date, measured mineral reference material date, the known reference material age, the $^{176}$Lu decay constant, and the uncertainty

associated with matrix fractionation correction. Uncertainties are reported at the 2SE level unless the quoted $p$ value is $< 0.05$, then the quoted uncertainty accounts for overdispersion following the method outlined in Vermeesch (2018).

## 2.2 U–Pb geochronology and trace elements

Monazites were analysed in situ by spot targeting guided by back-scattered electron (BSE) images collected on an FEI Quanta 450 Scanning Electron Microscope (SEM) housed at Adelaide Microscopy, The University of Adelaide. U–Pb and trace

element data were collected using a RESOlution-LR 193 nm excimer laser ablation system coupled to an Agilent 8900 ICP-MS/MS at Adelaide Microscopy, The University of Adelaide. Ablation was performed with a laser frequency of 5 Hz employing a 13 µm laser beam diameter with an average on-sample fluence of ~2.2 J cm$^{-2}$. The ablated sample material was transported from the laser cell to the ICP-MS by a He carrier gas (380 mL min$^{-1}$). Data acquisition consisted of (1) 30 seconds of baseline acquisition; (2) 30 seconds of continuous ablation, during which data were collected; and (3) ~25 seconds of

washout.

The isotopes collected were: $^{29}$Si, $^{31}$P, $^{43}$Ca, $^{89}$Y, $^{90}$Zr, $^{139}$La, $^{140}$Ce, $^{141}$Pr, $^{146}$Nd, $^{147}$Sm, $^{153}$Eu, $^{157}$Gd, $^{159}$Tb, $^{163}$Dy, $^{165}$Ho, $^{166}$Er, $^{169}$Tm, $^{172}$Yb, $^{175}$Lu, $^{202}$Hg, $^{204}$Pb, $^{206}$Pb, $^{207}$Pb, $^{208}$Pb, $^{232}$Th, and $^{238}$U. MAdel was used as the primary reference material to correct for elemental fractionation and mass bias (Payne et al., 2008), with 94-222 as a secondary reference material to monitor

precision and accuracy (Maidment, 2005). Standards were analysed after every 10–15 unknown analyses. For trace element concentrations, NIST 610 (Pearce et al., 1997) was analysed after every 10–15 unknown analyses. U–Pb isotope and trace element data were reduced using the 'U–Pb Geochronology' and 'Trace Elements' data reduction schemes in Iolite version 4.9.3 (Paton et al., 2011), respectively. Trace element data were calibrated using NIST 610. The internal standard element used was Ce and trace elements quantified by normalising wt% oxides to 100% totals. Error propagation and uncertainty reporting

follow the same approach discussed for Lu–Hf data. Secondary reference materials yielded results comparable to published values, with 94-222 yielding a weighted mean $^{206}$Pb/$^{238}$U age of $447 \pm 1$ Ma ($n = 41$, MSWD = 1.1, $p = 0.36$), within 2SE uncertainty of the reference age of $450.2 \pm 3.4$ Ma (Maidment, 2005). Weighted means and concordia plots were generated using IsoplotR (Vermeesch, 2018).

# 3 In situ Lu–Hf geochronology of candidate monazite reference materials

## 3.1 RW-1

Ten ~1 mm fragments of RW-1 mounted in a 25 mm epoxy resin disk were analysed in this study. The crystals are reddish-brown in colour and free of inclusions and cracks. RW-1 is a high-Th monazite that originates from a 20–30 m wide pegmatite dyke located in the Landsverk 1 quarry in the Evje-Iveland district, south Norway (Ling et al., 2017). Ling et al. (2017) presents

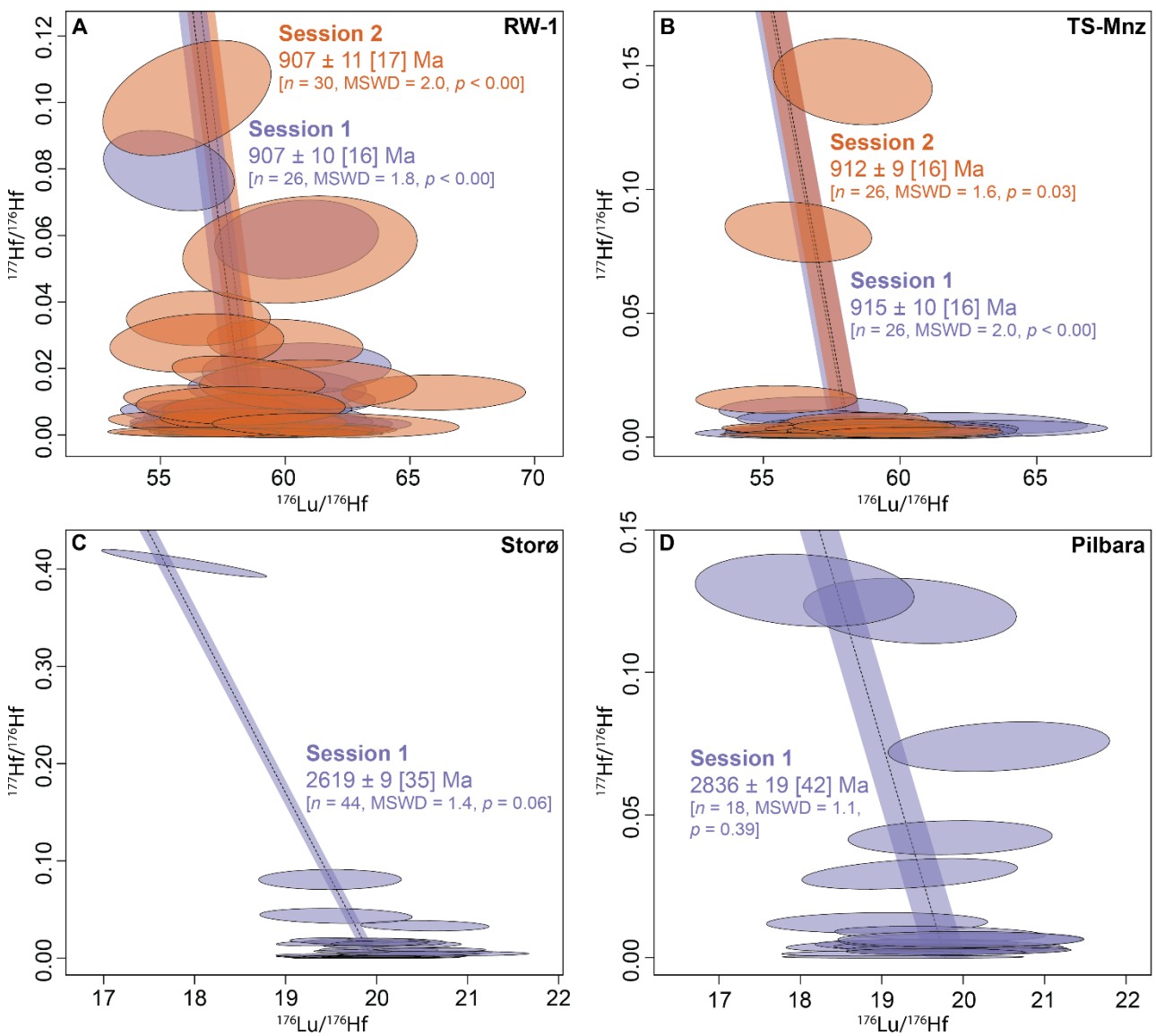

**Figure 1: Inverse isochron plots for (A) RW-1, (B) TS-Mnz, (C) Storø, and (D) Pilbara. Purple-coloured ellipses correspond to analyses from session 1, and orange-coloured ellipses correspond to analyses from session 2. Individual data-point uncertainties are 2SE.**

U–Pb ID-TIMS/ID-MC-ICP-MS isotopic data. These authors recommend the mean $^{207}Pb/^{235}U$ age of 904.15 ± 0.26 Ma (95%

conf.) as the best estimate for the crystallization age of the pegmatite hosting the RW-1 monazite (Ling et al., 2017). EPMA compositional data show that RW-1 has a $Ce_2O_3$ content of 25.22 wt%, $Nd_2O_3$ of 14.47 wt%, $ThO_2$ of 13.5 wt%, $Y_2O_3$ of 2.44 wt%, and $UO_2$ of 0.30 wt% (Ling et al., 2017). Additional LA-ICP-MS data show a Lu content of 27 ± 5 (2σ) ppm (Ling et al., 2017), making the sample amenable to in situ Lu–Hf geochronology.

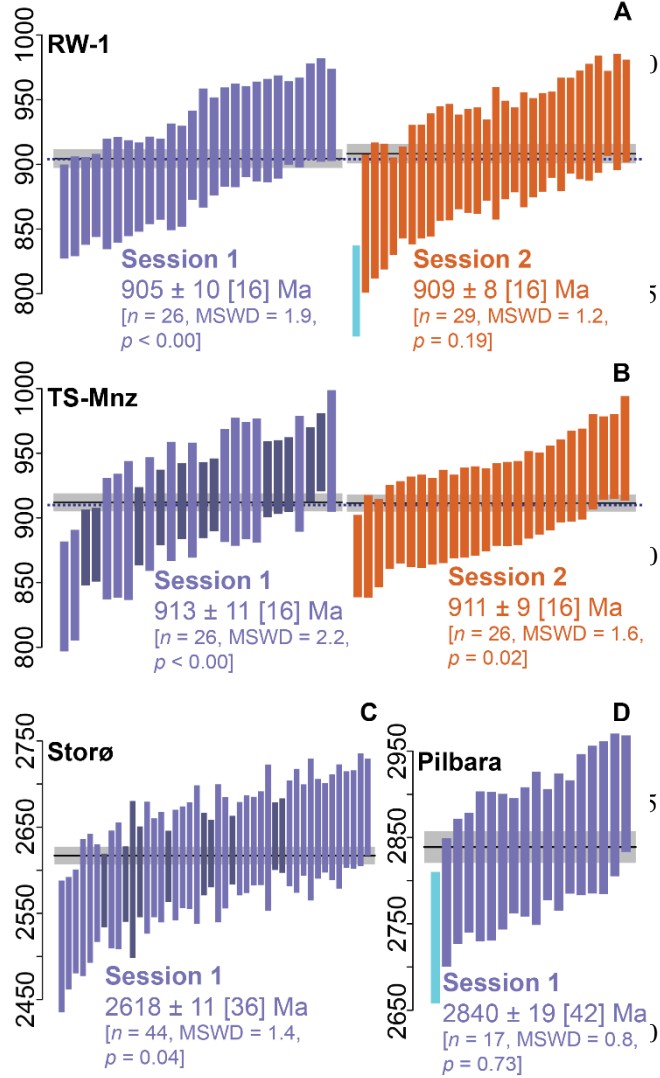

**Figure 2: Single-spot weighted mean plots for (A) RW-1, (B) TS-Mnz, (C) Storø, and (D) Pilbara. Purple-coloured bars correspond to analyses from session 1, and orange-coloured bars correspond to analyses from session 2. For TS-Mnz and Storø, darker coloured bars correspond to analyses collected using a 67 μm spot size, while lighter coloured bars were collected using a 43 μm spot. Individual data-point uncertainties are 2SE.**

In this study, RW-1 yields inverse Lu–Hf isochron dates of 907 ± 10 [16] Ma (n = 26, MSWD = 1.8, p < 0.00) and 907 ± 11 [17] Ma (n = 30, MSWD = 2.0, p < 0.00) and single-spot weighted mean dates of 905 ± 10 [16] Ma (n = 26, MSWD = 1.9, p < 0.00) and 909 ± 8 [16] Ma (n = 29, MSWD = 1.2, p = 0.19) for sessions 1 and 2, respectively (Fig 1A, 2A). Common Hf is very low, with most analyses yielding $^{177}Hf/^{176}Hf$ < 0.05. These dates are within uncertainty of published U–Pb ID-TIMS/ID-MC-ICP-MS ages (Ling et al., 2017).

### 3.2 TS-Mnz

A single ~7 mm fragment of TS-Mnz mounted in a 25 mm epoxy resin disk was analysed in this study. The crystal is reddish-brown in colour with abundant cracks that host thorite inclusions. These cracks were avoided during analysis, with only fresh monazite being analysed. The crystal, originally attained from a mineral dealer, likely originates from the Arendal region of Norway (Budzyń et al., 2021). U–Pb ID-TIMS data yields a $^{207}Pb/^{235}U$ age of 910.42 ± 0.34 Ma (2σ) (Budzyń et al., 2021). EPMA shows that TS-Mnz has a $Ce_2O_3$ content of 25.09 wt%, $Nd_2O_3$ of 15.92 wt%, $ThO_2$ of 4.80–9.44 wt%, $Y_2O_3$ of 2.83 wt%, and $UO_2$ of 0.16–0.29 wt% (Budzyń et al., 2021). LA-ICP-MS data also presented in Budzyń et al. (2021) indicates that TS-Mnz has a Lu content of 28.2 ± 3.7 (2σ) ppm, making the sample amenable to in situ Lu–Hf geochronology.

In this study, TS-Mnz yields inverse Lu–Hf isochron dates of $915 \pm 10$ [16] Ma ($n = 26$, MSWD = 2.0, $p < 0.00$) and $912 \pm 9$ [16] Ma ($n = 26$, MSWD = 1.6, $p = 0.03$) and single-spot weighted mean dates of $913 \pm 11$ [16] Ma ($n = 26$, MSWD = 2.2, $p < 0.00$) and $911 \pm 9$ [16] Ma ($n = 26$, MSWD = 1.6, $p = 0.02$) for sessions 1 and 2, respectively (Fig 1B, 2B). In session 1, the laser beam diameter was varied between 43 µm and 67 µm, however, aside from comparatively smaller uncertainties on analyses collected with the larger 67 µm spot size, no discernible difference in the accuracy of calculated Lu–Hf dates was observed (Fig. 2B). Common Hf is very low, with most analyses yielding $^{177}Hf/^{176}Hf < 0.05$. These dates are within uncertainty of published U–Pb ID-TIMS ages (Budzyń et al., 2021).

### 3.3 Storø

Thirty-five monazite grains ranging from 30 to 170 µm mounted on two epoxy resin disks were analysed in this study. The grains are yellow in colour with few cracks and inclusions. This sample originates from the Storø quartzite in West Greenland (Gardiner et al., 2023). Existing laser ablation split-stream ICP-MS data yield concordant U–Pb monazite dates between 2600 and 2630 Ma with an overdispersed concordia age of $2619 \pm 8$ Ma; the authors estimate the crystallisation age of monazite in this sample to be c. 2620 Ma (Gardiner et al., 2023).

In this study, Storø yields an inverse Lu–Hf isochron date of $2619 \pm 9$ [35] Ma ($n = 44$, MSWD = 1.4, $p = 0.06$; Fig. 1C) and a single-spot weighted mean date of $2618 \pm 11$ [36] Ma ($n = 44$, MSWD = 1.4, $p = 0.04$; Fig. 2C). The laser beam diameter was varied between 43 µm and 67 µm on a subset of analyses from Storø. Similar to the data from TS-Mnz in session 1, there was no discernible difference in the accuracy of calculated Lu–Hf dates observed between the two spot sizes (Fig. 2C). Common Hf is very low, with most analyses yielding $^{177}Hf/^{176}Hf < 0.05$. These dates are within uncertainty of the published U–Pb LA-SS-ICP-MS age of $2619 \pm 8$ Ma (Gardiner et al., 2023).

### 3.4 Pilbara

Ten ~1 mm monazite fragments mounted in a 25 mm epoxy resin disk were analysed in this study. The grains are reddish-brown in colour with few cracks and inclusions. Originating from a granitoid suite in the Pilbara Craton, Western Australia, this sample belongs to the Mawson Collection housed at the University of Adelaide. U–Pb dating of this sample yields an approximate age of c. 2870 Ma (unpublished).

Pilbara yields an inverse Lu–Hf isochron date of $2836 \pm 19$ [42] Ma ($n = 19$, MSWD = 1.1, $p = 0.39$; Fig. 1D) and a single-spot weighted mean date of $2840 \pm 19$ [42] Ma ($n = 17$, MSWD = 0.8, $p = 0.73$; Fig. 2D). Common Hf is very low, with most analyses yielding $^{177}Hf/^{176}Hf < 0.05$.

## 4 Comparing U–Pb and Lu–Hf data from the Arkaroola region

### 4.1 Geological background

The Arkaroola region of the northern Flinders Ranges, South Australia, hosts some of the highest heat producing basement rocks exposed at Earth's surface (De Vries Van Leeuwen et al., 2021; McLaren et al., 2006). These basement rocks comprise Mesoproterozoic granitoids and metasedimentary rocks which are exposed in two inliers, the Mount Painter and Mount Babbage inliers (Fig. 3). Overlying these high heat-producing basement rocks is a 12–15 km succession of sedimentary rocks, which form the Adelaidean stratigraphy of the Adelaide Superbasin (Lloyd et al., 2020; Paul et al., 1999; Preiss, 2000). Deposition of these sedimentary rocks began in the early-to-mid Neoproterozoic and terminated in the early Cambrian (Lloyd et al., 2020; Preiss, 2000).

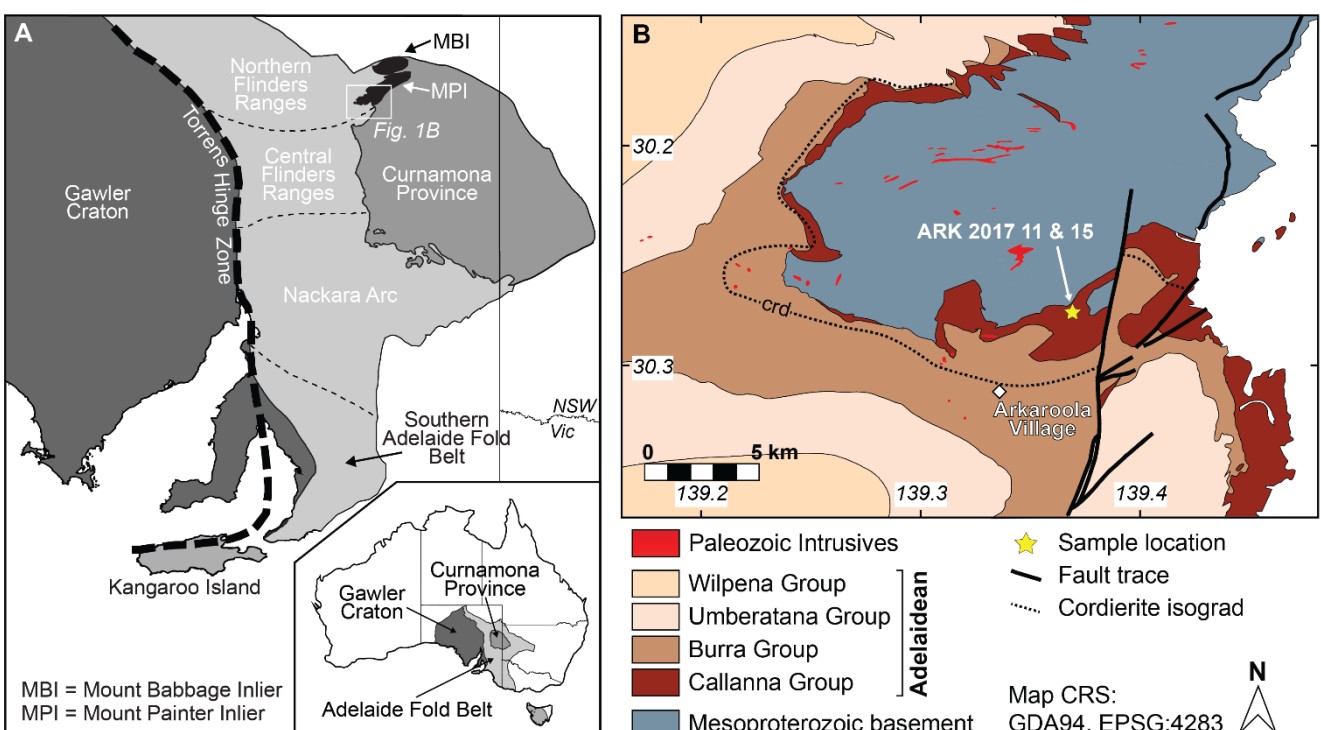

**Figure 3: (A) Geological setting of the Arkaroola region in South Australia, Australia; (B) Simplified geological map of the Arkaroola region.**

The accumulation of this thick sedimentary package on high heat producing basement rocks lead to the development of steep thermal gradients, resulting in high-temperature, low-pressure metamorphism of the basal portion of the sedimentary succession (De Vries Van Leeuwen et al., 2021; McLaren et al., 2006). This is recorded by the development of cordierite–biotite-bearing assemblages in metapelitic rocks (Fig. 3; De Vries Van Leeuwen et al., 2021; Mildren and Sandiford, 1995).

The consequence of this style of high heat production-driven 'burial' metamorphism, is that high thermal gradient conditions will persist providing the rocks are sufficiently deep.

    The cordierite-bearing metapelitic rocks at the base of the Adelaidean stratigraphy at Arkaroola record two distinct periods of monazite growth at c. 580–540 Ma and c. 450–400 Ma (De Vries Van Leeuwen et al., 2021), which are interpreted to reflect

the timing of thermally and hydrothermally catalysed monazite growth. The c. 580–540 Ma monazite population corresponds to a significant interval of subsidence and sedimentation in the Adelaide Superbasin, where ~5 km of sediment was deposited between c. 580–520 Ma (Paul et al., 1999; Preiss, 2000), significantly increasing the burial depth of the basement. Sediment accumulation was associated with the formation of progressively younger monazite ages for incipient metamorphism up stratigraphy (De Vries Van Leeuwen et al., 2021). The second monazite population at c. 450–400 Ma is more enigmatic, as

sedimentation in the Adelaide Superbasin had terminated by the onset of the c. 520–490 Ma Delamerian Orogeny (e.g., Foden et al., 2006; Preiss, 2000). However, evidence exists for a significant, regionally widespread hydrothermal-magmatic event between c. 460–400 Ma (e.g., Elburg et al., 2013; McLaren et al., 2006). Monazite also exhibits increasing HREE+Y contents between the c. 580–540 Ma and c. 450–400 Ma populations (De Vries Van Leeuwen et al., 2021), suggesting the thermal maxima was attained at c. 400 Ma, supporting the notion that increasing temperatures were a function of increasing burial

depth over at least a ~150 Myr period (De Vries Van Leeuwen et al., 2021).

### 4.2 Sample descriptions

    Two metapelitic samples, ARK 2017-11 and ARK 2017-15, were collected from the Paralana Quartzite, which forms the basal portion of the Adelaidean stratigraphy and occupies the unconformable interface with the high heat producing Mesoproterozoic basement rocks of the Mount Painter Inlier (Fig. 3). These samples, previously described in De Vries Van Leeuwen et al.

(2021), were derived from discrete metapelitic layers within broadly psammitic to quartzitic packages of the Paralana Quartzite. Although mineral modes vary between these two samples, both are mineralogically similar, exhibiting large porphyroblasts (up to 1 cm) wrapped by a strong fabric defined by biotite, plagioclase, and minor quartz. These porphyroblasts comprise fine-grained intergrowths of plagioclase and K-feldspar, biotite, and hematite, and are interpreted to represent altered cordierite.

    Monazite in these samples predominantly exists as large (up to ~500 μm) foliation-parallel elongate grains, anhedral grains throughout the matrix, or as inclusions within altered cordierite porphyroblasts. BSE images reveal two distinct generations of monazite (Fig. 4). The first generation, $mnz_1$, form dark BSE response poikiloblastic cores, containing rounded inclusions of quartz and rare hematite (Fig. 4). $Mnz_1$ often exhibits chaotic zoning patterns with high-Th monazite intergrowths, with

some grains in sample ARK 2017-11 also exhibiting patchy zoning with no clear core-rim relationship (Fig. 4L). The second generation, $mnz_2$, form as brighter BSE response rims mantling $mnz_1$, or grains with no core-rim relationships (Fig. 4). These

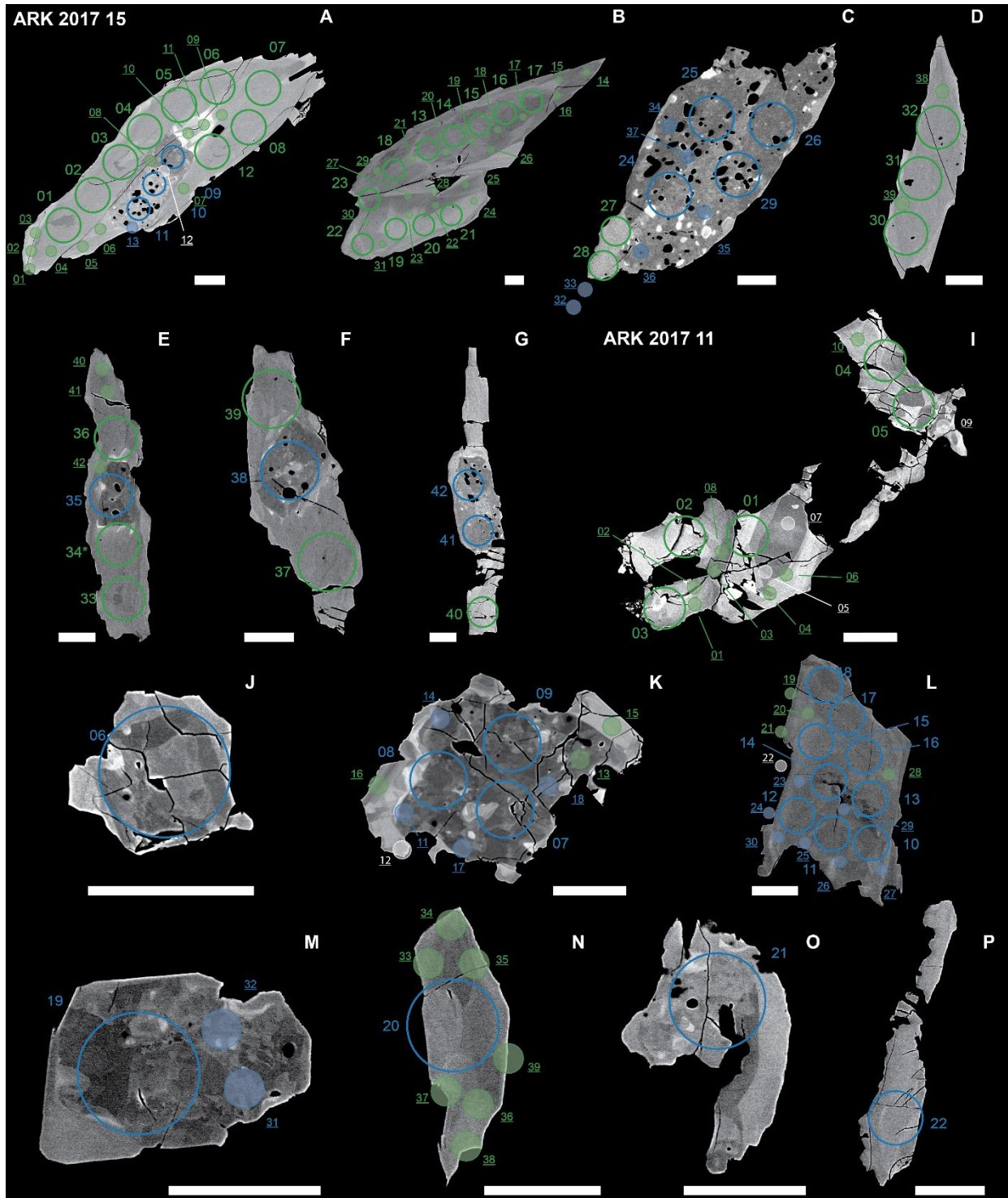

**Figure 4: BSE images of monazite grains from samples ARK 2017 15 and ARK 2017 11. Lu–Hf spots are represented by solid-lined circles and are coloured according to their microstructural domains (mnz₁ = blue, mnz₂ = green), corresponding to the colour scheme in Figure 5. U–Pb spots are represented by dash-lined shaded circles and are coloured according to the corresponding age populations shown in Figure 6. White-coloured dashed circles correspond to U–Pb analyses that display isotopic mixing. U–Pb spot numbers are in a smaller font size and underlined. Grains were re-polished between Lu–Hf and U–Pb analyses, as such, some U–Pb spots were placed beyond the extent of these BSE images.**

domains are inclusion-poor and can be homogeneous or exhibit patchy or wispy zoning patterns (Fig. 4). All analysed grains exhibit embayed margins.

### 4.3 In situ U–Pb and Lu–Hf geochronology

A total of forty-two U–Pb spot analyses were collected from sample ARK 2017-15, 7 of which belong to $mnz_1$, 34 belong to $mnz_2$ (Fig. 5A). An additional analysis, which yields a concordant $^{206}Pb/^{238}U$ date of 487 ± 13 Ma, is interpreted to reflect isotopic mixing between $mnz_1$ and $mnz_2$ domains (Fig. 5A). Chondrite-normalised REE data help to delineate data from $mnz_1$ and $mnz_2$ domains, with monazite belonging to the $mnz_2$ population consistently showing elevated HREE contents (Fig. 5B). Given the large spread of dates in both the $mnz_1$ and $mnz_2$ populations, the range of dates within each population is the preferred method of assigning an 'age' to each population. However, given that overdispersed dates often reflect underlying processes, we present them on Figure 5 for completeness. Analyses from $mnz_1$ yield $^{206}Pb/^{238}U$ dates of 604–571 Ma whereas those from $mnz_2$ are spread between 444 Ma and 390 Ma (Fig. 5A). These dates replicate the previously published monazite U–Pb data

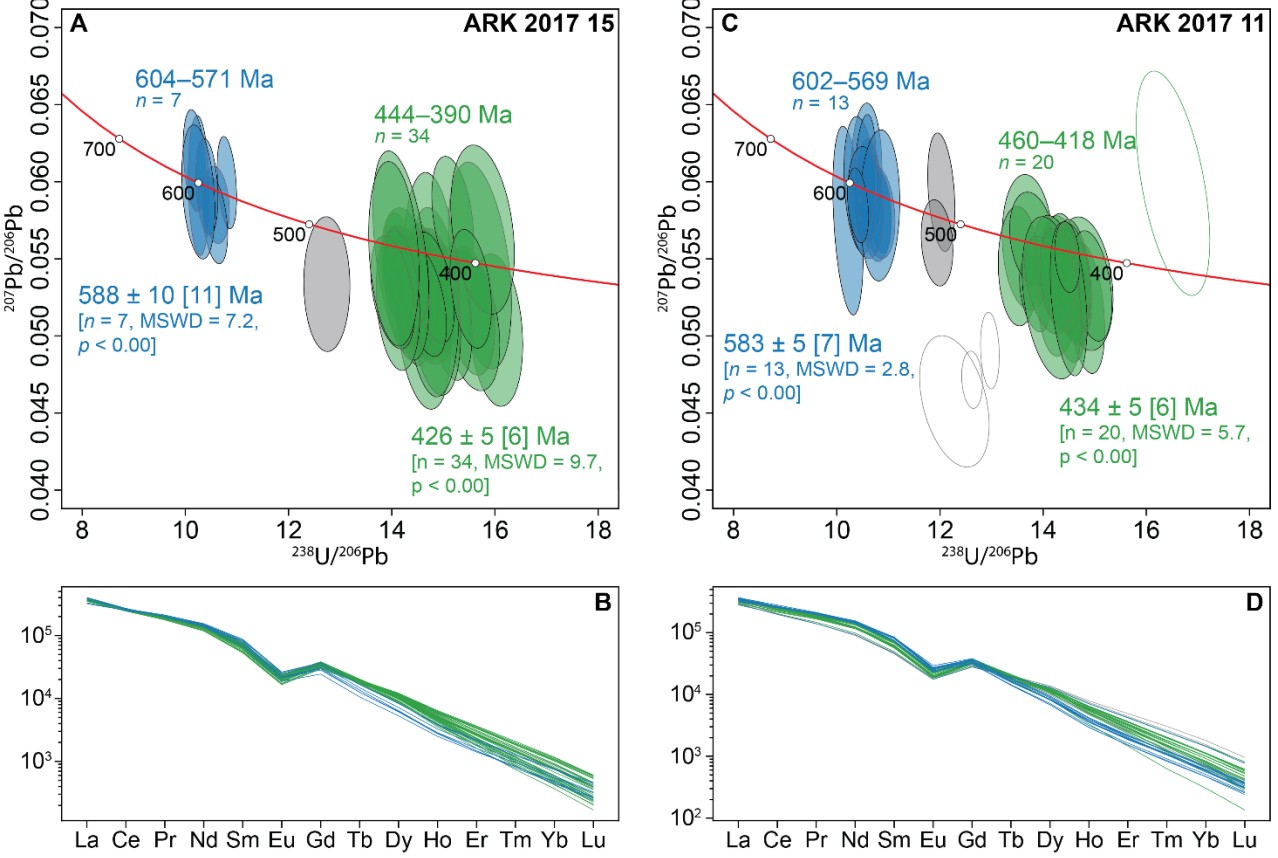

**Figure 5: (A, C) Tera-Wasserburg concordia plots for U–Pb analyses from samples (A) ARK 2017 15 and (C) ARK 2017 11; (B, D) Chondrite-normalised REE plots for analyses from samples (B) ARK 2017 15 and (D) ARK 2017 11. Blue ellipses and lines correspond to analyses from $mnz_1$ and green ellipses and lines correspond to analyses from $mnz_2$. Grey and unfilled ellipses in panels (A) and (C) and grey lines in panels (B) and (D) represent isotopically mixed analyses or erroneous analyses. Individual data-point uncertainties are 2SE.**

presented in De Vries Van Leeuwen et al. (2021). From these same monazite grains, 42 Lu–Hf spot analyses were collected, of which 11 were from $mnz_1$ domains and 31 were from $mnz_2$ domains (Fig. 6A). Two analyses from $mnz_1$ and one analysis from $mnz_2$ showed signs of isotopic mixing between the two domains and were not further considered for age calculations (Fig. 6A). Chondrite-normalised REE data from these analyses agree with that attained from U–Pb analyses, with $mnz_2$ analyses exhibiting elevated HREE contents (Fig. 6B). Analyses from $mnz_1$ yield an inverse Lu–Hf isochron age of $601 \pm 47$ [48] Ma (Fig. 6A; $n = 9$, MSWD = 1.3, $p = 0.21$) whereas analyses from $mnz_2$ yield and inverse Lu–Hf isochron age of $441 \pm$ 11 [13] Ma (Fig. 6A; $n = 30$, MSWD = 2.1, $p < 0.00$).

Thirty-nine U–Pb spot analyses were collected from sample ARK 2017-11, 13 of which belong to $mnz_1$ and 21 belong to $mnz_2$ (Fig. 5C). Five analyses yield intermediate $^{206}Pb/^{238}U$ dates between 518 Ma and 478 Ma, with two yielding concordant dates of $515 \pm 9$ Ma and $518 \pm 10$ Ma (Fig. 5C). Chondrite-normalised REE data show similar patterns to those in sample ARK 2017-15, with analyses from $mnz_2$ domains exhibiting elevated HREE contents (Fig. 5D). These analyses, as in sample ARK

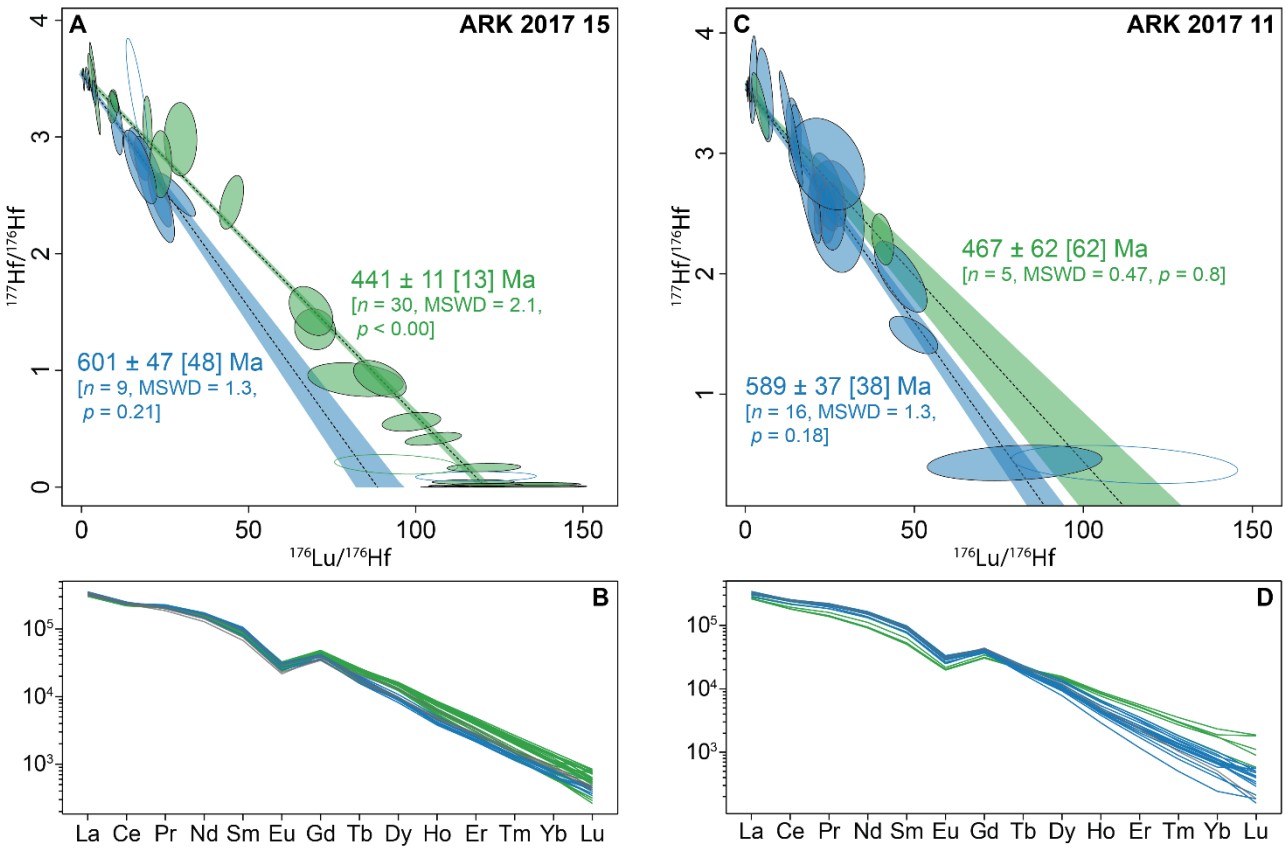

**Figure 6: (A, C)** Inverse isochron plots for samples **(A)** ARK 2017 15 and **(C)** ARK 2017 11; **(B, D)** Chondrite-normalised REE plots for analyses from samples **(B)** ARK 2017 15 and **(D)** ARK 2017 11. Blue ellipses and lines correspond to analyses from $mnz_1$ and green ellipses and lines correspond to analyses from $mnz_2$. Unfilled ellipses in panels (A) and (C) and grey lines in panels (B) and (D) represent mixed analyses which weren't considered for age calculations. Individual data-point uncertainties are 2SE.

2017-15, are considered to represent mixing between $mnz_1$ and $mnz_2$. Analyses from $mnz_1$ yield $^{206}Pb/^{238}U$ dates of 602–569 Ma whereas those from $mnz_2$ are spread from 460–418 Ma (Fig. 5C). A single analysis from $mnz_2$ yields an anomalously young $^{206}Pb/^{238}U$ date of 379 ± 13 Ma (Fig. 5C). Similar to sample ARK 2017-15, these data accurately replicate the monazite U–Pb data presented in De Vries Van Leeuwen et al. (2021). From the same grains, twenty-two Lu–Hf spot analyses were

collected from sample ARK 2017-11 of which 17 were from $mnz_1$ domains and 5 were from $mnz_2$ domains (Fig. 6C). Chondrite-normalised REE data from these analyses agree with that attained from U–Pb analyses, with $mnz_2$ analyses exhibiting elevated HREE contents compared to those from $mnz_1$ (Fig. 6D). A single mixed analysis from $mnz_1$ was excluded from age calculations (Fig. 6C). Analyses from $mnz_1$ yielded an inverse Lu–Hf isochron age of 589 ± 37 [38] Ma (Fig. 6C; *n* = 16, MSWD = 1.3, *p* = 0.18) while analyses from $mnz_2$ yielded and inverse Lu–Hf isochron of 467 ± 62 [62] Ma (Fig.

6C; *n* = 5, MSWD = 0.47, *p* = 0.80).

## 5 Discussion

### 5.1 Monazite reference materials

The two monazite reference materials with published ID-TIMS data investigated in this study, RW-1 and TS-Mnz, both yield inverse Lu–Hf isochron and weighted mean dates that lie within 2SE uncertainty and are accurate to <1 % of their published

U–Pb ages ( Fig. 1, 2; Budzyń et al., 2021; Ling et al., 2017). This demonstrates that the in situ Lu–Hf approach via LA-ICP-MS/MS, corrected for matrix-dependent fractionation to apatite reference materials, faithfully reproduces the published ID-TIMS/ ID-MC-ICP-MS U–Pb ages for monazite reference materials RW-1 and TS-Mnz (Budzyń et al., 2021; Ling et al., 2017).

Across two analytical sessions, RW-1 returned uncorrected inverse Lu–Hf isochron dates of 947 ± 11 Ma (*n* = 26, MSWD = 1.6, *p* = 0.02) and 949 ± 11 Ma (*n* = 30, MSWD = 1.9, *p* < 0.00) and TS-Mnz returned uncorrected inverse Lu–Hf isochron ages of 955 ± 11 Ma (*n* = 26, MSWD = 1.9, *p* < 0.00) and 955 ± 7 Ma (*n* = 26, MSWD = 1.4, *p* = 0.07), corresponding to apparent age offsets from their published ID-TIMS U–Pb ages of ~4.5–5.0 %. If these apparent age offsets are converted to matrix fractionation correction factors, RW-1 yields values of 4.78 ± 0.06 % and 5.00 ± 0.06 % while TS-Mnz yields values

of 4.94 ± 0.06 % and 4.94 ± 0.04 % for sessions 1 and 2, respectively. These values deviate slightly (< 1 %) from the matrix fractionation correction factors attained from apatite reference materials. However, the resulting < 0.5 % age difference for RW-1 and TS-Mnz when correcting to monazite versus apatite are within uncertainty. The similarity of matrix fractionation correction factors, along with the negligible common Hf contents and relatively high Lu contents (Budzyń et al., 2021; Ling et al., 2017), indicates that these reference monazites would be appropriate for calibrating unknown samples. Hence, although

RW-1 and TS-Mnz were used here as secondary reference materials, they can reliably be used to calibrate Lu–Hf ratios for matrix-dependant fractionation in future studies. In their recent study, Wu et al. (2024) also measured Lu–Hf ages for RW-1, but did not present the data, precluding a direct comparison between instruments and laboratories.

Although the Storø and Pilbara monazites are not as well-characterized as RW-1 and TS-Mnz, both yield inverse Lu–Hf isochron and weighted mean dates that fall within ~1% of their published U–Pb ages (Fig. 1, 2). This suggests they are suitable as secondary reference materials for evaluating the accuracy of post-acquisition calibrations and corrections (see above). Since Storø originates from a metasedimentary rock and Pilbara from a granitoid, it is evident that monazites from diverse rock types can serve as secondary reference materials, provided they meet the following criteria: (1) sufficient Lu content, (2) negligible common Hf, and (3) consistent Lu–Hf and U–Pb dates. In this regard, laboratories routinely performing in situ U–Pb monazite dating via LA-ICP-MS likely possess various in-house monazite reference materials that could also be used for Lu–Hf dating.

## 5.2 Comparing in situ U–Pb and Lu–Hf data from complex samples

In situ Lu–Hf geochronological data from samples ARK 2017-15 and 11 produces dates that lie within the spread of $^{206}Pb/^{238}U$ dates for both the mnz$_1$ and mnz$_2$ domains (c. 600–570 Ma and c. 460–390 Ma). This highlights that in situ Lu–Hf isotopic data attained via LA-ICP-MS/MS has the capacity to replicate ages attained via U–Pb LA-ICP-MS geochronology in geologically complex samples. Furthermore, it can resolve multiple age populations from samples which exhibit significant intragrain complexity, provided careful microstructural targeting is performed and companion trace element data are acquired.

U–Pb data from both mnz$_1$ and mnz$_2$ domains in both ARK samples exhibit large dispersion in concordant U–Pb dates. De Vries Van Leeuwen et al. (2021) argue that this dispersion corresponds to prolonged fluid-mediated dissolution-reprecipitation given the thermally energetic environment in which these rocks were metamorphosed. Although excess analytical scatter and/or common Hf incorporation cannot be ruled out, monazite dissolution-reprecipitation may also explain the overdispersion of the Lu–Hf dates for mnz$_2$ in sample ARK 2017-15 (MSWD = 2.1). This would suggest that (partial) dissolution of monazite effectively expels radiogenic Hf and its re-uptake during reprecipitation is limited. This in turn preserves the timing (and/or timespan) of fluid-rock interaction, behaving much the same as Pb during the same process (e.g., Harlov et al., 2011; Seydoux-Guillaume et al., 2002).

## 5.3 Applications and limitations

In situ Lu–Hf dating of monazite via LA-ICP-MS/MS presents an opportunity to gain geochronological information from a separate decay system to the commonly utilised U–Pb series. As demonstrated in this study, in situ Lu–Hf dating can aid in interpreting the ambiguous dispersion of U–Pb dates which is a well-documented feature of purportedly long-lived metamorphic systems (Clark et al., 2024; De Vries Van Leeuwen et al., 2021; Kirkland et al., 2016; Korhonen et al., 2013). It is debated whether this dispersion is due to low degrees of analytically unresolvable Pb-loss which generate seemingly protracted spreads of concordant dates, or if it represents a truly prolonged episode of metamorphism and/or fluid-rock interaction (e.g., Kirkland et al., 2016). However, when two separate isotopic systems both indicate dispersion (as seen in this study), it bolsters the argument that this represents a true geological signal. In addition to this, the Lu–Hf system in monazite

may be particularly useful in situations where the U–Pb system has been compromised or obfuscated by processes such as Pb-loss, non-radiogenic Pb incorporation, excess [206]Pb due to [230]Th uptake, or low U concentrations. Furthermore, the diffusivity of Lu and Hf in monazite is currently poorly understood. However, it is likely that that their behaviour differs from that of U, Th, and Pb. As such, the Lu–Hf system may have the ability to resolve geological processes that are not preserved (or are highly obscured) by the U–Pb system.

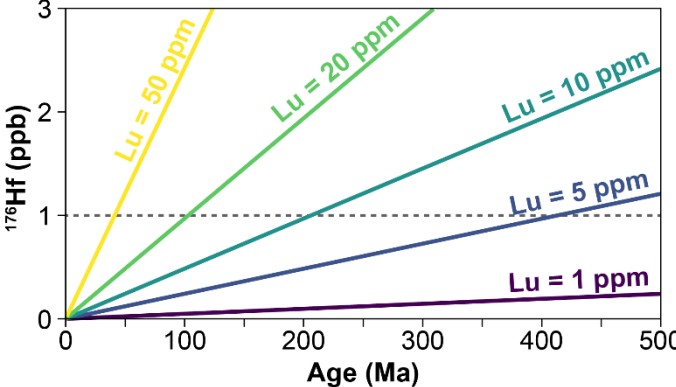

**Figure 7: Plot depicting the concentration of radiogenic [176]Hf accumulated for different Lu concentrations (1, 5, 10, 20, and 50 ppm) as a function of age. Dashed line indicates the approximate lower limit of detection on [176]Hf achieved in this study.**

At present, the greatest limiting factor for this technique is the analytical sensitivity of quadrupole instruments. Given the slow rate of radiogenic [176]Hf ingrowth, the accumulation of resolvable quantities is highly dependent on initial Lu concentrations (Fig. 7). Although monazite Lu concentrations are highly variable (dictated by numerous factors such as host rock/protolith composition, *P–T* conditions of metamorphism, magma/fluid chemistry etc.), they often fall in the range of 1 ppm to 50 ppm. Across the two analytical sessions conducted during this study, [176]Hf detection limits were typically observed to be ~1–3 ppb. From the data presented in Figure 7, it is evident that Lu-rich monazites (50 ppm) can resolve ages of ~50 Ma, while Lu-poor monazites (1 ppm) would take more than 2000 Myr reach [176]Hf concentrations at the lower limit of detection. This can be compensated for by increasing laser beam diameters (thus increasing count rates), albeit, at the expense of spatial resolution.

## 6 Conclusions

In situ Lu–Hf dating of monazite via LA-ICP-MS/MS faithfully reproduces published U–Pb ages of two monazite reference materials, RW-1 and TS-Mnz. We further demonstrate the approach for monazite from Arkaroola in South Australia, which formed during a complex and protracted geological history. These data replicate U–Pb geochronological data collected from the same grains and demonstrate that the Lu–Hf system within monazite is sensitive to resetting during fluid-mediated dissolution-reprecipitation. In situ Lu–Hf geochronology may find use in scenarios where the U–Pb system in monazite has been compromised (e.g., Pb-loss, common Pb contamination) and is unable to provide reasonable geological information.

## Supplement link

Supplementary Dataset 1: https://doi.org/10.25909/27441327.v3

## Author contribution

**ATDVVL:** Conceptualisation, Investigation, Writing - Original Draft, Visualisation **SG:** Conceptualisation, Investigation, Methodology, Writing - Review & Editing **MH:** Conceptualisation, Writing - Review & Editing **JM:** Resources, Writing - Review & Editing **SEG:** Methodology, Investigation, Writing - Review & Editing

## Competing interests

The authors declare that they have no conflicts of interest.

## Acknowledgements

The authors acknowledge the instruments and expertise of Microscopy Australia (ROR: 042mm0k03) at Adelaide Microscopy, University of Adelaide, enabled by NCRIS, university, and state government support. B. Wade and K. Neubauer from Adelaide Microscopy are thanked for their help with the collection of SEM data used in this study. S. Glorie is supported by an Australian Research Council Future Fellowship (FT210100906). J. Mulder is supported by Australian Research Council Fellowship DE24010128. K. Szilas and N. Gardiner are thanked for providing samples of the Storø Quartzite. R. Ickert is thanked for the editorial handling of this manuscript. N. Roberts and S. Walker are thanked for providing thoughtful and constructive reviews. This study was supported by the Mineral Exploration Cooperative Research Centre whose activities are funded by the Australian Government's Cooperative Research Centre Programme. This is MinEx CRC Document 2025/09.

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
