# Peer review of "A comparison between in situ monazite Lu–Hf and U–Pb geochronology"

_Geochronology, 2024_

## Author Comment (AC2)

**Reviewer RC2: Stephanie Walker**

This manuscript presents an important contribution by comparing in situ monazite Lu-Hf and U-Pb geochronology in a complex metamorphic terrane. Although the specific technique has already been outlined in a previous paper, this study is well-structured, methodologically rigorous, and effectively demonstrates the utility of Lu-Hf dating as an independent chronometer for validating U-Pb age interpretations. The authors provide a strong dataset with comprehensive analytical procedures. However, I have some concerns about the handling of the uncertainties which require further clarification.

**General comments**

Data processing: The matrix correction for Lu-Hf dating is based on apatite reference materials. While the justification is reasonable, the authors should explicitly state whether monazite-specific correction factors were tested and how any uncertainties from matrix mismatches were handled.

Apatite RMs yielded correction factors (CFs) of 4.40 ± 0.04 % and 4.71 ± 0.05 % for sessions 1 and 2, respectively. While monazite-corrected data are not presented here, the two monazite reference materials (RMs) yield correction factors of the same order as those from apatite RMs. We will expand this section to state the exact CF values for both RW-1 and TS-Mnz across both sessions, however, since the monazite correction factors

[Figure]

*Figure RC2-1: Inverse isochron plots for samples RW-1 (top panels) and TS-Mnz (bottom panels) demonstrating the use of monazite-based matrix fractionation correction factors (CF).*

are similar to those attained from apatite (i.e., deviating from the apatite derived CFs by < 1 %), we do not feel it necessary to present an additional dataset demonstrating their use. In Figure RC2-1, we present data from RW-1 and TS-Mnz which have had matrix fractionation corrections performed using CF values from monazite RMs (TS-Mnz used to correct RW-1, and RW-1 used to correct TS-Mnz). Although the calculated inverse isochron dates are within uncertainty of those corrected to apatite RMs, they provide slightly better accuracy in reproducing the published ID-TIMS ages of these RMs.

Matrix correction factors were not initially propagated onto the final inverse isochron/weighted mean dates (apologies for this oversight). This will be updated, and the propagated uncertainty values will be updated in the text and on Figures 1, 2, and 6.

Uncertainty propagation: Error propagation was undertaken involving quadratic addition of uncertainties from various sources (eg analytical session, reference material age, decay constant etc). However, there is no discussion of how systematic errors (instrumental drift and long-term reproducibility) were assessed.

Instrument drift was corrected for using LADR. Output uncertainties from LADR factor in intra-session instrument drift by fitting an 8th order polynomial spline (default fit in LADR) to NIST-610 SRM analyses which bracket the unknown analyses throughout the time-series data. The misfit of this calibration curve is accounted for in the exported uncertainties. For our comments on the long-term reproducibility of this method, we refer to the discussion on this topic in our response to RC1.

I'm guessing that the reference materials were processed in the same way as the unknowns? How do they compare over multiple sessions?

Yes, RMs are always processed in the same way as unknowns. Although two different apatite RMs were used between sessions 1 and 2 (Bamble-1 and OD-306, respectively), we refer to Glorie et al. (2024a) who present multi-session data for these RMs, demonstrating no meaningful variance across multiple sessions. We will also incorporate data from in-house secondary apatite standard HR-1 (long-term Lu–Hf age of 344 ± 2 Ma; Glorie et al., 2024a) to further appraise apatite corrections. HR-1 yielded corrected isochron ages of 348 ± 4 Ma (n = 15, MSWD = 1.30, p = 0.17) and 342 ± 3 Ma (n = 26, MSWD = 0.96, p = 0.52) for sessions 1 and 2, respectively.

Statistical handling of isochrons:

The study appropriately employs IsoplotR for isochron calculations, but the discussion lacks sufficient depth on the selection of anchored vs. unanchored regressions.

Unanchored isochrons give poor age results when Lu–Hf ratios are highly radiogenic (as in this study). The terrestrial variation in initial Hf ratios is miniscule on the scale of the isochron plot. However, anchoring the isochron to the terrestrial initial ratio, with an uncertainty that covers the entire range of terrestrial variation, prevents obtaining isochron results with impossible initial ratios. The choice of anchor, within the initial range makes no difference to the final isochron age (see discussion below). Some additional discussion around this point will be added to the main text.

The authors use a fixed initial $^{177}Hf/^{176}Hf$ of 3.55 ± 0.05 for isochron regressions. While this may be appropriate, it introduces a level of model dependence that should be discussed in more detail. Were alternative initial ratios tested?

Because $^{177}Hf/^{176}Hf$ of all terrestrial sources covers only a very small range (encompassed by the uncertainty on 3.55 ± 0.05), we believe that anchoring the data within this range is a valid approach. Free regressions on data which exhibit little spread along the isochron can lead to spurious upper intercepts yielding geologically implausible initial Hf values (Vermeesch, 2024). This is particularly pertinent to monazite data, as monazite should contain no inherited Hf (nominally). As such, one would expect the data to be clustered around extremely radiogenic values, impeding the ability to fit a precise isochron via free regression. This is supported by the data we present from our analysis of various RM/SRM monazites, where there is significant clustering of analyses from samples towards radiogenic values. In Figure RC2-2, we demonstrate that anchoring the isochron to values

of $^{177}$Hf/$^{176}$Hf = 3.5 or $^{177}$Hf/$^{176}$Hf = 3.6 (upper and lower limits of the range we allowed our regressions to intercept) makes no statistical difference to the resultant dates (i.e., they are all within uncertainty).

[Figure]

*Figure RC2-2: Inverse isochron plots for samples RW-1 (top panels) and ARK 2017 15 (bottom panels) anchored at values of $^{177}$Hf/$^{176}$Hf = 3.5 (left panels) and $^{177}$Hf/$^{176}$Hf = 3.6 (right panels).*

Some of the inverse isochrons have MSWD values greater than 2. The manuscript states that this suggests prolonged fluid-rock interaction, but alternative explanations (e.g. analytical scatter, common Hf incorporation) should also be considered.

We agree that some discussion about alternative explanations for the observed excess scatter is warranted. Although we believe that the scatter is likely due to fluid-rock interaction (supported by the U–Pb dataset), we cannot unequivocally rule out other options. We will add additional text to the manuscript discussing this.

**Minor comments**

Figures: The figures are clear and well-labelled, but the colour scheme in Fig 4 for the different microstructural domains could be more distinct to improve readability.

The format of Figure 4 will be modified to improve visibility.

References: These are comprehensive, but there are some inconsistencies in the formatting such as missing DOIs for some references.

DOIs will be added to all references (where available).

Supplementary data: Why are there such an absurd number of decimal places for the ratios? Obviously this depends on the precision of the values, but I imagine no more than two or three are actually significant.

The number of decimal places will be reduced.

**Recommendation**

While the manuscript presents valuable and well-executed research, improvements in uncertainty handling and statistical interpretation are required prior to publication. Once the above comments are addressed, I am confident that this will be a robust foundation for future monazite studies.

**References cited in this response**

Glorie, S., Hand, M., Mulder, J., Simpson, A., Emo, R. B., Kamber, B., Fernie, N., Nixon, A., and Gilbert, S.: Robust laser ablation Lu–Hf dating of apatite: an empirical evaluation, Geological Society, London, Special Publications, 537, 165–184, https://doi.org/10.1144/SP537-2022-205, 2024a.

Vermeesch, P.: Errorchrons and anchored isochrons in IsoplotR, Geochronology, 6, 397–407, https://doi.org/10.5194/gchron-6-397-2024, 2024.

---

## Author Response (AR1)

**Reviewer 1: Nick Roberts**

Nice paper, well presented, with a useful and clear case study.

There is already a paper on this method. This has been cited, and uses different instrumentation. The case study described here is a nice addition.

There is no comment on the fact that older monazites will be easier to date (in terms of measurable radiogenic Hf that is). How young will this method be useful for using this instrumentation? That is concentration specific of course, but clearly unpicking the Alpine-Himalayan orogen is not going to be easy.

Although monazite Lu concentrations are highly variable and are dictated by a plethora of factors (i.e., host rock/protolith composition, P-T conditions of metamorphism, magma/fluid chemistry etc.), they often fall in the range of 1 ppm to 50 ppm. In Figure R1, we provide a plot visualising the concentration of ingrown radiogenic 176Hf between 0 Ma and 500 Ma with total Lu concentrations varying between 1–50 ppm. As expected, increasing Lu concentrations in monazite allow for younger ages to be resolved, and as Nick mentions, older monazites are easier to date at lower Lu concentrations given the longer radiogenic ingrowth times.

Across the two analytical sessions conducted during this study,  $^{176}$ Hf detection limits were typically observed to be ~1–3 ppb. In Figure R1 below, we can see that monazite with a relatively high Lu concentration of 20 ppm would take c. 100 Myr to accumulate enough radiogenic Hf to reach the lower limit of detection achieved in this study. As such, this method, barring exceptionally Lu-rich monazite, would struggle to date Cretaceous samples or younger. It may be possible to push this to slightly younger ages by employing larger spot sizes (and thus increased sensitivity) but this would require exceptionally large monazite grains to analyse.

A new Discussion section (5.3 Applications and limitations) has been added to the main text which summarises much of the above discussion (Lines 321–350). The analytical detection limits of 175Lu, 176Hf and 178Hf have also been added to Supplementary Dataset 1.

*Figure R1: Plot depicting the concentration of radiogenic* 176*Hf accumulated for different Lu concentrations (1, 5, 10, 20, and 50 ppm) as a function of age.*

It is a shame that the study does not include any of the most commonly used monazite RMs, e.g. Stern, 44069, Manangotry, Moacyr/Bananeira.

We agree that it would have been ideal to include these reference materials (RMs), unfortunately our lab does not currently have these RMs available for analysis, and as such, we do not know if they contain sufficient Lu and low 177Hf to be useful as a Lu–Hf RM. We tried several common RMs, such as 222 and MAdel, and these did not qualify as suitable RMs. However, we believe that the two monazite RMs analysed here with established ID-TIMS ages are sufficient to show that the Lu–Hf system in monazite provides accurate and geologically meaningful age information.

Error propagation: at this stage, this is all that can be done, and covers the basics of the calculations. A comment on the fact that long-term reproducibility is not accounted for, and may also be a large contributor to the total age uncertainty would be prudent.

Given that this is a nascent technique, it is not feasible to accurately constrain the long-term reproducibility of this method. However, the two monazite RMs, RW-1 and TS-Mnz, analysed across two analytical sessions in this study yield combined isochron dates of 906.8  $\pm$  7.4 Ma (RSD = 0.82 %) and 913.4  $\pm$  6.6 Ma (RSD = 0.72 %), respectively. Furthermore, Glorie et al. (2024b) reports a combined isochron date from 6 analytical sessions of 930.3  $\pm$  1.4 Ma (RSD = 0.15 %). These data may point towards the method (in situ Lu–Hf dating via LA-ICP-MS/MS more broadly) having a long-term excess variance similar to, if not less than, conventional LA-ICP-MS techniques (e.g., Sliwinski et al., 2022).

The paper relies on its predecessors to describe common Hf, Yb corrections etc. I am not suggestion repetition, but perhaps comments on the key issues and important considerations would be useful.

A correction was performed for isobaric interference of  $^{(176+82)}$ Lu on  $^{(176+82)}$ Hf by monitoring  $^{(175+82)}$ Lu and subtracting a proportion of this signal from  $^{(176+82)}$ Hf based on the present-day  $^{176}$ Lu/ $^{175}$ Lu ratio (0.02659). No corrections were performed for isobaric interferences from  $^{(176+82)}$ Yb on  $^{(176+82)}$ Hf, as this has been demonstrated to be negligible (~0.00003 % of total measured  $^{172}$ Yb). A sentence summarising the above discussion has been added to the main text (Lines 64–69).

Figures – I didn't see it stated that bars/ellipses are 2sigma.

This has been added to the captions of Figures 1, 2, 5, and 6.

Data – The tables should comprise mass spectrometer signals for at least some of the measurements, as per widely shared recommendations for U-Pb. The decimal places are too many for the ratios.

Background-subtracted counts per second data for 176Lu (measured on 175 amu), 176Hf (measured on 258 amu), 177Hf (measured on 260 amu) has been added to Supplementary Dataset 1 to demonstrate instrument sensitivity.

The number of decimal places for isotope ratios has been reduced in Supplementary Dataset 1.

Line 30 - 'orogens' would be more accurate that 'terranes'

Orogens is now used in place of terranes (Line 31).

Line 44 – It is not clear how the two listed 'problems' with the approach of Wu et al, "hinder exploring the application of Lu-Hf monazite to its full potential".

Wu et al. (2024) show that their approach requires larger interference corrections, particularly on Yb, and the inability to accelerate ions decreases sensitivity. We demonstrate that we can obtain results at similar or better precision while utilising smaller laser beam diameters, without the need to perform cumbersome interference corrections. An explanation of this has been added to the main text (Lines 43–45).

Line 56 – Were spot sizes mixed during each session? Were they mixed between samples and RMs, and does this matter? If not, then this needs to be demonstrated. Different spot sizes will change the downhole fractionation patterns, but it is understandable on a quadropole instrument that the data may be too imprecise to measure any difference accurately.

Spot sizes were varied between 43 and 67 µm for TS-Mnz and Storo in Session 1. For the unknowns analysed in Session 2 (ARK 2017 11 & 15), a spot size of 67 µm was employed, except for instances where the size of the target domain was small, in which case a smaller 43 µm spot was employed to minimise mixing between domains. This is now outlined in the main text (Lines 56–57, 169–172, 183–184). Additionally, we now differentiate analyses that employed different spot sizes on the weighted mean plots presented in Figure 2. Given that individual ellipses are not easily discernible on the inverse isochron plots presented here (highly clustered data), analyses with differing spot sizes will not been differentiated (Figs 1 & 6). However, we will provide the spot size used for each analysis in Supplementary Dataset 1.

Based on our data, using a quadrupole mass spectrometer, there is no observable difference between the two spot sizes (aside from the smaller uncertainties on analyses employing a large spot size stemming from increased counts). No downhole fractionation corrections were applied, as there was no observable downhole fractionation. This is consistent with the results of Simpson et al. (2021), where no downhole fractionation was observed in garnet, apatite, or xenotime, using laser beam diameters between 43 µm and 120 µm.

Line 57 – The reality of this method, is that this spot size is commonly larger or similar to the total length of metamorphic monazites found in typical pelites metamorphosed at mid-crustal conditions. U-Th-Pb spots are typically 5 to 15 microns, which is why multiple domains can be dated from single grains.

We agree that this is a limitation of the method. However, given that monazite grain size in metamorphic rocks is dictated by numerous factors such as the rate of intergranular element transport in the presence (or absence) of fluid(s), element availability, and volume diffusion rates, it's not a simple "one size fits all" consideration. In our experience, providing the rocks of interest are not exceptionally fine-grained (e.g., hornfels), with some initial sample triaging (done prior to creating mounts/thin sections), locating numerous large monazite grains is feasible in most mid-crustal metapelites. We favour a method where large portions of rock are slabbed and then scanned using a  $\mu$ XRF spectrometer. Regions with elevated P that do not also exhibit elevated Ca are likely monazite grains (as opposed to apatite).

Line 74 – Indistinguishable – using this method and instrumentation. This does not mean that the ideology can be applied to all minerals, all instruments and all conditions. A matrix-matched approach should always be strived for, even if this is not possible at first. Non-matrix-matching allows for poor 'traceability' of the method.

It is of course possible that the quadrupole instrumentation employed here may be too imprecise to discern differences between matrix-matched and non-matrix-matched correction factors. However, this would require a separate study to systematically compare results from a quadrupole and a more precise technique (e.g., multi-collector ICP-MS). This statement, now on **Lines 75–77**, has been modified to acknowledge that this assertion is only true using a quadrupole ICP-MS.

Line 80 – Were trace elements checked against any published monazite data, or are they only considered to be non-quantitative? Were the Ce contents measured with EMPA, or just assumed? What Ce value was used if assumed?

The trace element data presented here are considered semi-quantitative and were not checked against published data. During data reduction, Ce contents for TS-Mnz and RW-1 were set to published values of 21.42 wt% and 21.39 wt%, respectively. For Storo, Pilbara, ARK 2017 11, and ARK 2017 15, Ce contents were set to 20 wt%. These details have been added to the main text (Lines 90–91). In the manuscript trace element data is only relied upon to discern different monazite domains in samples ARK 2017 11 and ARK 2017 15, as such, semi-quantitative data is deemed adequate.

Line 268 - I personally would not call these established – but they have published ID U-Pb data, and that is the point.

This is a fair comment. We have removed the suggestion that these are established RMs and simply state that these are monazite RMs with published ID-TIMS data (Line 278).

Line 269 – Perhaps also list the accuracy in terms of %, i.e. "accurate to <1%".

This is a great suggestion. We now state the accuracy of the Lu–Hf dates from RW-1 and TS-Mnz presented in this study with respect to their published ID-TIMS U–Th–Pb dates (Line 279).

Line 288 – True, but many labs only work with Phanerozoic monazite RMs – and it is unclear how the accuracy will degrade with younger and younger samples/RMs with this method.

We refer to our response to the first comment. This is a function of Lu concentration. Monazite with higher Lu concentrations will be dateable to younger ages. Exceptionally Lu-rich monazite (Lu > 50 ppm) should, in theory, be able to resolve ages younger than ~50 Ma using the instrumental setup outlined in this study.

Line 295 – Yes, indeed. I would also make the point that the companion trace element data are critical to unpick multiple populations.

We agree that coupling in situ Lu–Hf isotopic data with trace element geochemistry is a necessity when collecting and interpreting data from complex samples such as those in this study. This is now mentioned in the main text (Lines 310–311).

Line 300 - missing a "to"

Added a "to" (Line 314).

**References cited in this response**

Glorie, S., Simpson, A., Gilbert, S. E., Hand, M., and Müller, A. B.: Testing the reproducibility of in situ Lu–Hf dating using Lu-rich garnet from the Tørdal pegmatites, southern Norway, Chemical Geology, 653, 122038, https://doi.org/10.1016/j.chemgeo.2024.122038, 2024b.

Simpson, A., Gilbert, S., Tamblyn, R., Hand, M., Spandler, C., Gillespie, J., Nixon, A., and Glorie, S.: In-situ LuHf geochronology of garnet, apatite and xenotime by LA ICP MS/MS, Chemical Geology, 577, 120299, https://doi.org/10.1016/j.chemgeo.2021.120299, 2021.

Sliwinski, J. T., Guillong, M., Horstwood, M. S. A., and Bachmann, O.: Quantifying Long-Term Reproducibility of Zircon Reference Materials by U-Pb LA-ICP-MS Dating, Geostandards and Geoanalytical Research, 46, 401–409, https://doi.org/10.1111/ggr.12442, 2022.

**Reviewer 2: Stephanie Walker**

This manuscript presents an important contribution by comparing in situ monazite Lu-Hf and U-Pb geochronology in a complex metamorphic terrane. Although the specific technique has already been outlined in a previous paper, this study is well-structured, methodologically rigorous, and effectively demonstrates the utility of Lu-Hf dating as an independent chronometer for validating U-Pb age interpretations. The authors provide a strong dataset with comprehensive analytical procedures. However, I have some concerns about the handling of the uncertainties which require further clarification.

**General comments**

Data processing: The matrix correction for Lu-Hf dating is based on apatite reference materials. While the justification is reasonable, the authors should explicitly state whether monazite-specific correction factors were tested and how any uncertainties from matrix mismatches were handled.

Apatite RMs yielded correction factors (CFs) of  $4.40 \pm 0.04$  % and  $4.71 \pm 0.05$  % for sessions 1 and 2, respectively. While monazite-corrected data are not presented here, the two monazite RMs yield correction factors of the same order as those from apatite RMs. We have expanded this section to state the exact CF values for both RW-1 and TS-Mnz across both sessions (Lines 288–293), however, since the monazite correction factors are similar to those attained from apatite (i.e., deviating from the apatite derived CFs by